# Case Study: Applying Decision Focused Learning in the Real World

**Shresth Verma**
Google Research India
vermashresth@google.com

**Aditya Mate**
Harvard University
aditya_mate@g.harvard.edu

**Kai Wang**
Harvard University
kaiwang@g.harvard.edu

**Aparna Taneja**
Google Research India
aparnataneja@google.com

**Milind Tambe**
Google Research India
milindtambe@google.com

## Abstract

Many real world optimization problems with underlying unknown model parameters are solved using the predict-then-optimize framework. In particular, a model is learnt to first predict the parameters of the optimization problem, which is subsequently solved using an optimization algorithm. However, this approach maximises for the predictive accuracy rather than the quality of the final solution. Decision Focused Learning (DFL) solves this objective mismatch by integrating the optimization problem in the learning pipeline. Previous works have only shown the applicability of DFL in simulation settings. In our work, we consider the optimization problem of scheduling limited live service calls in Maternal and Child Health Awareness Programs and model it using Restless Multi-Armed Bandits (RMAB). In collaboration with an NGO, we conduct a large-scale field study consisting of 9000 beneficiaries for 6 weeks and track key engagement metrics in a mobile health awareness program. To the best of our knowledge this is the first real world study involving Decision Focused Learning. We demonstrate that beneficiaries in the DFL group experience statistically significant reductions in cumulative engagement drop, while those in the Predict-then-Optimize group do not. This establishes the practicality of use of decision focused learning for real world problems. We also demonstrate that DFL learns a better decision boundary between the RMAB actions, and strategically predicts parameters which contribute most to the final decision outcome.

## 1 Introduction

Predict-then-optimize [7, 4, 5] is a framework that is often used in scenarios where an optimization problem is to be solved but the parameters defining the optimization problem are unknown. This is a *two-stage approach*: The first stage is to learn a predictive model which maps from some environment features to the parameters. Subsequently, in the second stage, the optimization problem formulated using the predicted parameters is solved. However, there is a key shortcoming in this two-stage framework. While the mapping function maximizes for the predictive accuracy of parameters, we are interested in the solution quality of the optimization problem parameterized by the predicted parameters. Decision-Focused Learning (DFL) [3, 29, 14, 25] is proposed to address this mismatch between the training objective and the evaluation objective by embedding the optimization problem within the training pipeline. However, until now, Decision Focused Learning has only been studied through simulated experiments.

2022 Trustworthy and Socially Responsible Machine Learning (TSRML 2022) co-located with NeurIPS 2022.

In our work, we consider the optimization problem of constrained sequential resource allocation solved using Restless Multi-Armed Bandits (RMAB). An RMAB consists of N independent stochastic processes (arms), and a planner which decides to pull or not pull each of the arms. The Whittle index heuristic [28] is the dominant approach for solving RMABs and is shown to be asymptotically optimal [27]. However, for computing Whittle Indices, transition dynamics of each arm must be known. When arm features are correlated with transition dynamics, historical data on arm pulls is leveraged to learn a mapping from arm features to transition dynamics. The learnt mapping function is then used to predict the unknown parameters for new arms and solve the subsequent optimization problem.

In this paper, we present the first work showcasing the real-world impact of DFL for RMABs through a large-scale field study. For conducting the field study, we collaborate with an NGO in India working in mobile health space for maternal and child health awareness. Extensive cell phone coverage in India allows non-profits to feasibly reach underserved communities for delivering health information via text or voice messages [19, 12]. Unfortunately, such programs suffer from a dwindling engagement over time, with a large number of beneficiaries dropping out from the program. NGOs often make use of health workers to personally reach out to beneficiaries through service calls, encourage their participation and address complaints. However, health workers' availability and time are scarce resources; only a limited number of beneficiaries can be given a service call and it is thus crucial to optimize which beneficiaries receive these personal service calls.

In prior works, a RMAB model using the previously mentioned two-stage learning approach has been used for optimizing service call scheduling in the field [16]. We compare this two-stage approach with a DFL approach in optimizing service calls. Engagement is a key metric that captures beneficiaries' participation in the mobile program. Our results show that allocating health worker resources using a DFL policy reduces drop in engagement by $31\%$ as compared to the no-service call baseline. On the other hand, the benefit from TS policy is not statistically significant. We also show that live service calls made by health care workers using DFL policy have higher effectiveness than TS policy resulting in better short-term and long-term outcomes in listenership behavior.

Furthermore, in this work, we perform detailed post-hoc analysis to explain how DFL is making decisions and why those decisions result in a better performance. Our novel contributions are as follows:

- We show results from the first large-scale field study of Decision Focused Learning being applied to maternal and child health domain.
- We show that by optimizing for decision quality rather than predictive accuracy, DFL results in statistically significant improvement in final decision quality measured through engagement metric in the mobile health program.
- We show through a series of detailed analysis how DFL is more strategic in optimizing for the final decision quality.

Our positive results thus pave the way for future works applying Decision Focused Learning in real world optimization problems with underlying unknown problem parameters.

## 2 Related Works

The predict-then-optimization [6] framework (or two-stage learning) solves for an optimization problem with unknown parameters by learning a predictive model of parameters from environment features and subsequently solving the optimization problem. However, this two-stage process separates out the prediction and optimization problems, thereby causing a mismatch between the predictive loss that is minimized and the evaluation metric that is desired to be maximized [10, 13, 11]. Decision Focused Learning [29, 14, 5], solves this problem by embedding the optimization problem as a differentiable layer in a deep learning pipeline. In [3, 18] one-shot optimization problems such as stochastic programming and security games are solved in an end-to-end manner, while [24, 8] focus on solving sequential decision making problems using Decision Focused Learning.

In our work, we focus on the optimization problem of constrained sequential resource allocation solved using Restless Multi-Armed Bandits (RMAB). RMABs have been used in real world applications such as anti-poaching patrol planning [21], healthcare interventions [16, 15], planning anti-poaching patrols [21] and machine repair and maintenance [9]. The complexity of optimally solving RMAB

problems is known to be PSPACE hard [17]. Whittle Index approach [28] is an approximate solution to RMAB problem which is asymptotically optimal under the indexability condition [27, 1, 26]. However, for computing the Whittle Index, transition dynamics must be known. Under unknown system dynamics, [16, 22] leverage the predict-then-optimize framework for learning a predictive model of transition dynamics from features using historical data. [25] instead propose a Decision Focused learning methodology for Restless Multi Armed Bandit problems for generalized N-state MDP as well as a belief state MDP to optimize for decision quality. While the results are promising, they are only limited to simulated experiments.

## 3 Background

**ARMMAN**  ARMMAN is a non-governmental organization in India focused on reducing maternal and neonatal mortality among underprivileged communities. The NGO operates a mobile health service that disseminates preventive health information to expectant or new mothers (beneficiaries) on a weekly basis via automated voice messages. A large fraction ($\sim 90\%$) of mothers in the program are below the World Bank international poverty line [30] and the program has so far served over a million mothers. However, despite the success of the program, beneficiaries' engagement with the voice calls dwindles over time with $22\%$ of beneficiaries dropping out of the program within just 3 months of enrolment. Live Service calls made by health workers can encourage beneficiaries' participation. However, the health workers' availability is limited and thus, only a fixed number of live service calls can be made every week. This constraint necessitates a smart scheduling strategy of which beneficiaries to reach out every week to best utilize health workers' efforts.

**Restless Multi-Armed Bandits**  We consider the Restless Multi-Armed Bandit model with N independent arms each characterized by a 2-action Markov Decision Process (MDP). Each MDP is defined using the tuple $\{\mathcal{S}, \mathcal{A}, R, \mathcal{P}\}$ where $\mathcal{S}$ refers to the state space, $\mathcal{A}$ is the action space, which in our case is discrete and binary, $\mathcal{A} \in \{0, 1\}$. $R$ is the reward function such that $R : \mathcal{S} \times \mathcal{A} \times \mathcal{S} \mapsto \mathbb{R}$. $\mathcal{P}$ is the transition function, such that $\mathcal{P}(s, a, s')$, $(s, s') \in \mathcal{S}, a \in \mathcal{A}$ represents the probability of transitioning from state $s$ to $s'$ under action $a$. The policy function $\pi : \mathcal{S} \mapsto \mathcal{A}$ is defined as the mapping from states to action.

In our problem setup, we consider a 2-state 2-action MDP problem. Based on our discussions with the NGO, states are defined using the engagement metric. If a beneficiary listens to at least 1 call for more than 30 seconds in a week, they are said to be in Engaging state ($s = 1$). Otherwise, the beneficiary is in Non-Engaging state ($s = 0$). The timestep of the MDP is chosen to be a period of 1 week. The actions correspond to whether to deliver (active) or not deliver (passive) a live service call to a beneficiary. Additionally, the NGO can only deliver $K$ live service calls in a week. The reward function at any given timestep is defined to be same as the current state $R(s, a) = s$. The planner's goal is then to maximize expected long term reward (engagement). Starting from a state $s_0$, this is defined as $V_\gamma^\pi(s_0) = \mathbb{E}_{s_{t+1} \sim \mathcal{P}} \left[ \sum_{t=0}^{\infty} \gamma^t R(s_t, \pi(s_t), s_{t+1} | \pi, s_0) \right]$.

## 4 Training

**Data**  Prior to the start of field study, we have historical data available on beneficiaries' listenership behavour between January 2022 to May 2022. Specifically, we have state trajectories of 19000 (N) beneficiaries over a period of 5 weeks (T), along with the action chosen for every beneficiary at every timestep. Additionally, we have socio-demographic features for every beneficiary obtained at registration time. These features cover information such as age, gestational age, income, education, parity, gravidity, language of automated call, and registration channel.

**Learning TS Model**  In [16], TS model is shown to cut $\sim 28\%$ engagement drops as compared to a Round-Robin baseline. In our work, we consider outperforming the TS baseline to show applicability of the DFL model. Thus we follow a similar setup of the TS model as described in [16]. A mapping function $f$ is learnt that predicts the Transition Probabilities given the socio-demographic features $x_i$ for the $i_{th}$ arm. Predicted Transition Probabilities can then be obtained using $f(x^i), i \in [N]$. Since our problem domain consists of two states and two actions, we have to predict four transition probabilities. We model the mapping function as a neural network $f_w$ parameterized by the weights $w$. $f_w$ is designed using two fully connected layers followed by four outputs and finally logistic

function is applied to obtain probabilities. $f_w$ is learnt by minimizing the negative log-likelihood of observed trajectories $\mathcal{T}$ under the predicted transition probabilities $f(x)$. The loss function $\mathcal{L}$ is thus given by $\mathcal{L}(f_w(x), \mathcal{T}) = \mathbb{E}_{i \in [N]} -\log(\mathcal{T}^i | f_w(x^i))$. The weights $w$ of this neural network are optimized by backpropagating the gradient $\frac{d\mathcal{L}(f_w(x), \mathcal{T})}{dw}$.

**Learning DFL Model** We replicate the Decision Focused learning pipeline from [25]. Instead of optimizing for predictive accuracy, we optimize for the final decision outcome. Off-Policy Policy Evaluation (OPE) is used to quantify the decision outcome from a learnt policy given the past trajectories from a different policy. The DFL architecture uses the same predictive model as TS described in the previous section. However, once Transition Probabilities are predicted, we compute Whittle Indices $WI$ which parameterize a differentiable policy. Finally, the differentiable evaluation objective is formulated using OPE of learnt policy $\pi$. The weights of the predictive model are learnt by maximizing the final objective and backpropagating through the complete pipeline. The gradient is thus given by $\frac{d \, \text{OPE}(\pi^{WI}, \mathcal{T})}{dw}$. We refer the reader to [25], for more details on DFL pipeline.

## 5 Field Trial

In this section, we describe the real-world study [1] conducted to evaluate the performance of Decision Focused Learning policy against the Two-Stage baseline. As mentioned previously, in earlier work, the two-stage approach was shown to outperform other non-ML baselines [16], and therefore outperforming the two-stage approach is important to show the utility of DFL.

We consider a fixed cohort of 9000 beneficiaries registered between the months of April 2022 to June 2022. We split these set of beneficiaries into three groups - (1) DFL group where beneficiaries are chosen for live service calls according to the Whittle Index Policy learnt using Decision Focused learning, (ii) TS group where beneficiaries are chosen for live service calls according to the Whittle Index Policy learnt using Two-Stage learning, and (iii) Current Standard of care (CSOC group), where no live service calls are delivered to the beneficiaries. Additionally, we make sure that the distribution of socio-demographic features and start-state are the same across the three groups. Beneficiaries become eligible for live service calls 2 months post their enrolment into the program. Within each experiment group, we choose $K = 300$ beneficiries for live service call every week based on NGO's constraints. These live service calls are sent out weekly for a period of 6 weeks. We continue to monitor the cohort for 4 more weeks even after the study ends to measure the sustained effect of live service calls.

## 6 Real World Impact

In this section, we showcase the results from the field study. We also define multiple evaluation metrics and demonstrate how the different policies fare against each other.

**Weekly and Cumulative Engagement** We first present the results from our study using the Engagement Metrics proposed in previous field study for our domain [16]. Engagement at time $t$ for the $i^{th}$ beneficiary, represented by $E^i(t)$ is defined as 1, if the beneficiary listens to at least one automated call in a week for more than 30 seconds and 0 otherwise. Since the engagement of beneficiaries dwindles over time, we can measure the drop in engagement relative to the engagement at start. The engagement drop is defined as $E^i_{drop}(t) := E^i(0) - E^i(t)$ and cumulative engagement is given by $E^i_{cumulative\_drop}(t) := \sum_{\zeta=0}^{\zeta=t} E^i_{drop}(\zeta)$. The cumulative engagement drop prevented over the CSOC group is simply the difference in cumulative engagement drop of the policy and the CSOC group. Figure 1 shows the cumulative engagement drops prevented over CSOC group for DFL and TS policies. We see that DFL prevented more drops across all weeks and by the end of study, DFL group has **560** more engagement drops prevented over the CSOC group as compared to TS group which only prevents **181** engagement drops. Given a total of 1765 cumulative engagement drops in the CSOC group, DFL group has 31% fewer cumulative engagement drops as compared to CSOC group while TS only results in 10% reduction in cumulative engagement drops.

---

[1] All experiments reported in this paper are approved by an ethics review board at the NGO

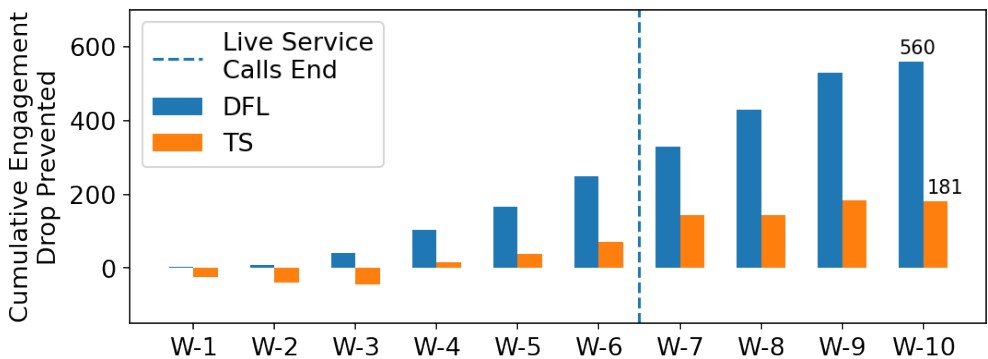

Figure 1: Weekly Cumulative Engagement Drop Prevented for the DFL and TS groups. Live service calls are only delivered for the first 6 weeks, after which, all three groups are only passively observed.

Table 1: Effectiveness of Live Service Call evaluated through multiple metrics

| Policy | Change in Mean Duration | Change in No. of Engagements | Change in E/C | Change in C/S |
|--------|------------------------|-----------------------------|---------------|---------------|
| DFL | 17.054 | 0.094 | 0.191 | 0.061 |
| TS | 6.764 | 0.009 | 0.077 | 0.013 |

**Statistical Significance** We also establish statistical significance of DFL's benefit using regression analysis [2]. We fit a linear regression model to predict the output variable $E^i_{cumulative\_drop}$ by giving beneficiary features $x_i$ as input along with and an indicator variable $T_i$ denoting whether a beneficiary belongs to DFL ($T_i = 1$) or CSOC ($T_i = 0$) group. The regression coefficient for $T$ is found to be $0.19$ with p-value of $0.024$. On the other hand, a similar comparison between TS vs CSOC yields a regression coefficient of $0.06$ for $T$ with p-value of $0.48$. Thus, belonging to the DFL group resulted in significantly positive impact on cumulative engagement drops while for TS, no such statistical significance could be established.

**Live Service Call Effectiveness** While the whittle index policy maximizes the reward, which is defined using the engagement metric, we also measure if the policy improved other metrics characterizing listenership. Thus, we define some metrics quantifying the change in listenership behavior of a beneficiary from the time window of $w$ days before to the time window of $w$ days before after a live service call. Within a given time window, we consider the following metrics:

1. Mean Duration: The mean duration of calls listened to within the time window.

2. No. of Engagements: The numbers of calls in the time window listened to more than 30 seconds.

3. Engagements to Connections (E/C) Ratio: The ratio of numbers of calls engaged with (engagements) to numbers of calls connected within the time window. A connected call is defined as the one listened to for more than 0 seconds.

4. Connections to Scheduled (C/S) Ratio: The ratio of numbers of calls connected to numbers of calls scheduled for a beneficiary within the time window.

Finally, we calculate the change in these metrics between the time window before and after a live service call. Table 1 reports the mean change in these metrics across the three experimental groups for a time window of 14 days. We observe that across all the metrics, DFL group has a much higher change in listenership behavior through live service calls as compared to the TS group. For instance, we can interpret the value of $17.054$ in Mean Duration metric for DFL as active actions in the DFL group resulting in beneficiaries listening to on average 17 seconds more of an automated call. This is in contrast to TS group, where live service calls only resulted in beneficiaries listening to 6 seconds more of an automated call. Using t-test for comparison of means, we find that for each of these metrics, mean change is statistically higher for the DFL group as compared to the TS group with p-value$< 0.05$.

Table 2: Different Evaluation metrics for DFL and TS policies

| Policy | Transition Probability Error Metrics | | | | Rank Metrics | |
| --- | --- | --- | --- | --- | --- | --- |
| | MAE All | MAE Top-K | Mean NLL All | Mean NLL Top-K | Precision @ K | Spearman's Correlation |
| DFL | 0.31 | **0.35** | 0.79 | **0.62** | **0.41** | **0.30** |
| TS | **0.25** | 0.37 | **0.42** | 0.69 | 0.22 | 0.179 |

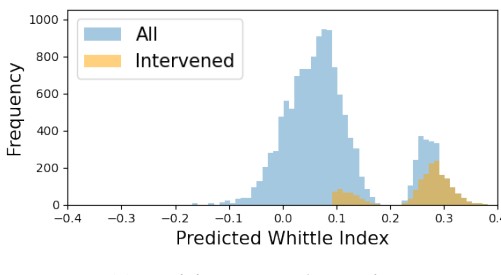

(a) Decision Focused Learning

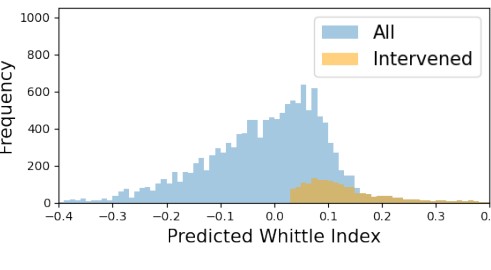

(b) Two Stage

Figure 2: Predicted Whittle Index Distribution and Beneficiaries Intervened for TS and DFL groups across all weeks.

## 7   Understanding DFL

The Decision Focused Learning method consists of an end-to-end pipeline starting from features to predicted Transition Probabilities to computed whittle index and finally the decision of whether a beneficiary is in top-K list chosen for live service call. In this section, we interpret the DFL's strategy in contrast with the Two-Stage policy by performing post-hoc analysis across all these steps.

As a first step for this analysis, we compute the ground truth transition probabilities using the observed trajectories of beneficiaries during the time period of field study. Once Ground Truth Transition Probabilities are estimated, we subsequently compute the Ground Truth Whittle Index and Ground Truth top-K ranks.

**Top-K Rank Lists**   We consider the ordered list of beneficiaries according to the predicted whittle index in the Two-Stage and DFL experiment groups. Additionally, the True Top-K rank list is also computed using the ground truth whittle index. To measure the agreement between the two lists, we use the following metrics:

1. Precision @ K: This metric counts the proportion of relevant beneficiaries in the top-K positions of the policy rank list and is widely used in classification [31, 20] and ranking problems [23]. The precision @ K in our problem is given by:

$$\text{Precision @ K} = \frac{(\text{No. of beneficiaries common in Policy Top-K list and True Top-K list})}{K}$$

2. Spearman's Rank Correlation: This metric calculates the rank correlation between the Predicted Whittle Index and Ground Truth Whittle Index of Policy's Top-K ranked beneficiaries.

In Table 2, we show the different rank metrics for the two comparison groups. In all the weeks, we find the DFL group has a higher agreement with the True Top-K ranks as compared to the Two-Stage experiment group.

**Whittle Indices**   For beneficiaries belonging to each of the experimental groups, we have the corresponding computed Whittle Index from predicted Transition Probabilities. We call it the Predicted Whittle Index (although these values are not directly predicted by the Neural Network models). Figure 2 shows the distribution of Predicted Whittle Index for DFL and TS experiment groups in Blue. We also mark the beneficiaries who are chosen for Active action within each experimental group in orange.

A striking observation is that within the DFL group, the whittle indices have a bimodal distribution as opposed to a unimodal distribution for the Two-Stage group. This suggests that in DFL, the model

is trying to learn a decision boundary between the beneficiaries to deliver active and passive action. This contrasts with the Two-Stage model where the objective is solely to learn accurate transition probabilities.

**Predicted Transition Probabilities**   Given the ground truth and predicted transition probabilities for both DFL and TS policies, we compute for the whole population (i) the Mean Negative Log Likelihood (NLL) of observed trajectories under predicted transition probabilities and (ii) the prediction error using Mean Absolute Error (MAE). In Table 2, we show that DFL has both higher MAE and higher Mean NLL as compared to TS. Thus DFL model is poorer in predicting the transition probabilities. However, if we compute these metrics for just the true top-K beneficiaries (MAE Top-K and Mean NLL Top-K), we find that DFL has lower MAE as well as Mean NLL than TS. This suggests that the DFL focuses on correctly predicting the transition probabilities for beneficiaries who will actually be picked, in contrast to the TS, which optimizes for predictive performance for the whole population. It must be noted, that the predictive performance of TS is impacted due to limited historical data around active actions (limited service calls made by the NGO).

## 8   Conclusion

We conduct the first large-scale real world study of Decision Focused learning through an RMAB problem in maternal and child health domain. This is in contrast to previous works which only show applicability of DFL in simulation settings. We conclude that Decision Focused Learning by optimizing for decision quality results in higher participation of beneficiaries in the program. In particular, DFL policy results in $31\%$ reduction in engagement drops as compared to a no-service call baseline(Figure 1). DFL's strategic selection of actions also results in more effective live service calls as demonstrated through significant change in average duration of calls listened to and the number of calls listened to for more than 30 seconds (Table 1).

From the analysis showcased in previous sections, we attribute the success of DFL to the following:
(i) The predicted whittle index distribution from DFL policy is bimodal in contrast to a unimodal distribution in TS (see Figure 2) indicating that DFL model learns a decision boundary to highly rank beneficiaries that would benefit significantly more from receiving the service call than the rest of the population.
(ii) DFL is more aligned with the optimal policy as shown by a higher rank correlation with the True Top-K Beneficiaries as compared to TS (Table 2).
(iii) While TS results in a lower predictive error for the population as a whole, DFL by optimizing for decision quality results in improved transition probability prediction for the top-K beneficiaries (Table 2).

## 9   Ethics

The study design and all the experiments were approved by ethics board at the NGO and the research group. Both written and digital consent is received from beneficiaries to subscribe to the automated and live service calls. Additionally, all data is owned and maintained by the NGO while the research group only accesses an anonymized version of the data. Lastly, domain experts from the NGO were deeply involved in all phases of the model iteration and design process.

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
