# OpenReview forum: "Case Study: Applying Decision Focused Learning in the Real World"
_NeurIPS.cc/2022/Workshop/TSRML — TSRML2022_

### Official Review · Reviewer_mRZB · 2022-10-13

**Overall Rating:** 7

**Summary:**

The authors conduct a real-world study of decision focused learning (DFL) in the maternal health domain. DFL is a methodology for solving optimization problems with unknown parameters. In such settings, some parameters of the optimization are unknown, but may be estimated from data. In order to maximize the quality of the solution (as opposed to superficial quantities like predictive accuracy), DFL embeds the optimization problem as a differentiable layer in the learning problem. In this work, the authors replicate an existing DFL pipeline and apply it to the problem of reducing maternal and neonatal mortality among underprivileged communities in India. Specifically, they learn the parameters of a restless multi-armed bandit problem, which (1) models expectant/new mothers' engagement with calls used to convey preventive health information and (2) uses this model to decide which expectant/new mothers receive an automated message and which receive a live call. The authors measure  the drop in engagement relative to the engagement at the start, and find that DFL prevented more drops than the relevant baselines at a statistically significant level.

**Strengths:**

To the best of my knowledge, this paper is the first to show the benefits of DFL in a real-world field study. The study appears to be well-designed and proper ethical considerations appear to have been taken, which is important given the nature of the test domain (providing health information to expectant/new mothers). Additionally, the results of the field study are significant, as they show that DFL reduces engagement drops by 24% compared to a natural machine learning baseline.

**Weaknesses:**

I see no major weaknesses of the work. One minor critique is that I believe more effort should be made to convince the reader that using DFL or other ML-based methods is the "right thing to do" for this problem. In particular, the authors say that the ML baseline they compare to was shown to outperform other non-ML baselines in a previous work ([16]), but no mention of what these baselines are, or why they are relevant is mentioned in this paper. It might be good to include a comparison and justification for such baselines in an appendix.

**Overall Recommendation:**

Overall, this work examines a real-world case study of DFL in the maternal health domain. Thematically it is a good fit for the workshop and the findings are significant, so I recommend acceptance.

**Review Confidence:**

3: The reviewer is fairly confident that the evaluation is correct

---

### Official Review · Reviewer_FUTU · 2022-10-21
**A well-designed field study of a simple but important application of decision focused learning**

**Overall Rating:** 8

**Summary:**

The paper describes a randomized controlled comparison of optimal placement of phone calls to optimize engagement in a medical application between two methods: (a) a baseline two-stage model which first trains a model optimized for accuracy of predicting response and then learns the decision policy based on the previous model, and (b) a model which is directly optimizing the response-prediction model for expected long-term award through. The study found that the direct optimization ("decision focused learning") yielded better aggregate engagement. Finally, the paper compared the policies.

**Strengths:**

- Interesting and well-design study.
- Mostly well-written paper.


**Weaknesses:**

- The technical details are unclear without the full version in [25].
- It is unclear but would be interesting to understand why DFL policy outperforms the two-stage one. The current rationale seems to be that DFL learns a decision boundary, but why is that happening in the first place. In other words, how to characterize the disparity between prediction-focused objective + policy evaluation and the policy evaluation as the objective.

**Overall Recommendation:**

The paper is a good fit for the workshop. I think it describes a field study, rare in machine learning, makes it even more valuable for this community.

**Review Confidence:**

3: The reviewer is fairly confident that the evaluation is correct

---

### Decision · Program_Chairs · 2022-10-23

**Decision:**

Accept

**Comment:**

Very nice study on decision focused learning with Field Trials.